# Optimising Calls to Large Language Models with Uncertainty-Based Two-Tier Selection

**Guillem Ramírez**
University of Edinburgh
gramirez@ed.ac.uk

**Alexandra Birch**
University of Edinburgh
A.Birch@ed.ac.uk

**Ivan Titov**
University of Edinburgh
University of Amsterdam
ititov@inf.ed.ac.uk

## Abstract

Researchers and practitioners operating on a limited budget face the cost-performance trade-off dilemma. The challenging decision often centers on whether to use a large LLM with better performance or a smaller one with reduced costs. This has motivated recent research in the optimisation of LLM calls. Either a *cascading* strategy is used, where a smaller LLM or both are called sequentially, or a *routing* strategy is used, where only one model is ever called. Both scenarios are dependent on a decision criterion which is typically implemented by an extra neural model. In this work, we propose a simpler solution; we use only the uncertainty of the generations of the small LLM as the decision criterion. We compare our approach with both cascading and routing strategies using three different pairs of pre-trained small and large LLMs, on nine different tasks and against approaches that require an additional neural model. Our experiments reveal this simple solution optimally balances cost and performance, outperforming existing methods on 25 out of 27 experimental setups.

## 1 Introduction

Large Language Models (LLMs) offer a high performance for a wide range of text tasks. Their widespread popularity both in research and industrial applications necessitates an understanding on how to optimally use them. Bigger models tend to have better performance, while smaller models are faster and cheaper to run. Deciding which model to use is a common dilemma for many researchers and practitioners with limited budgets, time-constraints or environmental concerns.

Recent works attempt to optimise calls to a set of LLMs. In this work we consider the set-up with two LLMs, where one is more expensive with greater performance than the other. In this scenario, there are two main strategies (Figure 1): *routing*, where a query from a user is directed to only one model based on a decision criterion; *cascading*, where the query always goes to the cheaper model and may subsequently go to the more expensive model depending on the cheaper model's output. These previous studies use one of these calling strategies, and involve either using an auxiliary model to score an LLM output (Chen et al., 2023; Sakota et al., 2023; Ding et al., 2024; Madaan et al., 2023) or using repeated calls to the small cheaper LLM (Yue et al., 2024; Madaan et al., 2023).

Studies that use an auxiliary model introduce further complexity in the optimisation approach, and it remains unclear when practitioners should rely on these auxiliary models. Not only do they require additional training, but they also usually require specific training data and the auxiliary models may not generalise to other tasks. For studies that rely on repeated calls to the small LLM, this approach can become expensive, undermining the original practical motivation for its use. It is perhaps for these reasons, that neither approach has gained traction among practitioners.

However, we question whether these additional models or the repeated calls are required to optimise LLM calls. We hypothesise that they may be unnecessary in many use cases, as we can extract confidence measures from the generations of the small model. To investigate this,

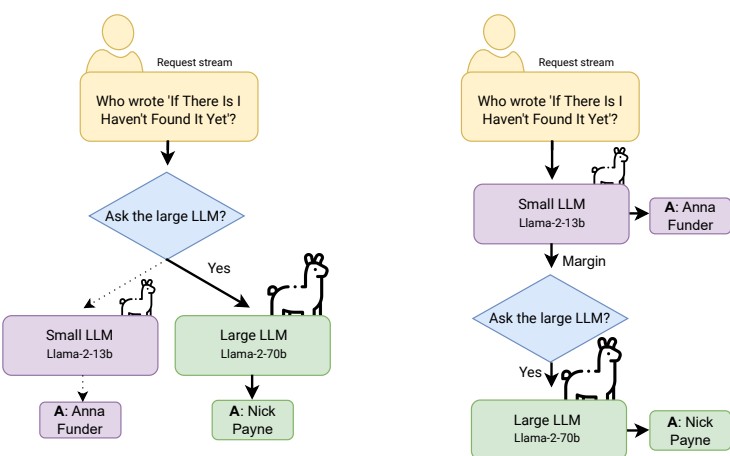

Figure 1: Routing (left) attempts to select the LLM with the best cost-accuracy trade-off given an incoming query. In cascading (right), all queries are passed through the small model, and depending on its output, the large LLM is consulted. We propose using a cascading approach that uses the margin of the generations to score outputs from the small LLM.

we propose a *cascading* policy that uses a simple measure of confidence of the small LLM to decide whether the large LLM needs to be called. We evaluate our policy on nine tasks, and for three different pairs of pre-trained small and large LLMs. We focus specifically on short-generation tasks, which are widely used and versatile, and do not include long-generation tasks in this study due to the additional complexity they introduce for evaluation. In our experiments on classification, multiple-choice and Question-Answering (QA) tasks, this policy outperforms methods that require additional training.

We believe our findings suggest that the optimisation of LLM calls should gravitate towards understanding their existing available signals, as opposed to training and running further auxiliary models. Moreover, simple and cheap policies have a lower entry cost for researchers and practitioners, which has an impact on their adoption.[1]

The key contributions of this work are as follows:

- We propose using the margin of the generations of LLMs to optimise LLM calls. The approach is simple, non-parametric, and does not require data or running multiple calls to the small model.
- We test our policy on nine tasks with three different pairs of small and large LLMs, and against relevant neural methods that use auxiliary models. We find that Margin Sampling outperforms the other methods, despite using fewer resources. In addition, we further test our policy in a multi-task set-up and obtain similar results.
- Our results underscore the importance of understanding signals from LLMs in the context of optimising calls to them, in contrast with previous work, which require additional training of auxiliary models.

## 2   Related work

**LLM uncertainty**   The margin between the two most likely classes has been widely regarded as an uncertainty measure adopted from early Active Learning literature (Scheffer et al., 2001; Luo et al., 2004) and considered more recently in the context of Knowledge Distillation from LLMs (Baykal et al., 2023; Ramírez et al., 2023). Other measures of uncertainty exist for LLMs (Baan et al., 2023; Huang et al., 2023). The primary challenge in

---

[1]We make our code publicly available: https://github.com/guillemram97/margin_llms

text generation lies in the difficulty of distinguishing between various forms of uncertainty. Specifically, when evaluating the confidence level of a text generator, we would ideally want to concentrate on uncertainties related to the change in meaning. This entails distinguishing uncertainties that affect the intended meaning from meaning-preserving variations in the generated text. In this study, we address this challenge by concentrating on tasks that require generating very short sequences of tokens. We find that even a basic approach to uncertainty can be advantageous. Additionally, while it might seem logical to assess uncertainty through multiple text outputs from a model, this would lead to significant computational costs, which are impractical for our purposes. Therefore, we employ a straightforward method that does not necessitate generating multiple text samples.

**Optimisation of inference costs**   Several methods have been proposed to improve the latency of LLMs, such as speculative decoding (Leviathan et al., 2023), knowledge distillation (Bucila et al., 2006; Hinton et al., 2015) and model quantisation (Jacob et al., 2018). However, operational costs rather than latency are the focus of this work. Our method only requires access to the logits of a small pre-trained LLM with no need for its parameters.

**Optimisation of LLM API Calls**   Recent work deals with the problem of optimising calls to a pool of LLMs (Wang et al., 2024). Sakota et al. (2023) and Lu et al. (2023) propose to train an auxiliary model that predicts the success of calling each LLM. Similarly, Ding et al. (2024) developed a model that predicts the quantitative benefit of utilizing a smaller language model over a larger one.

Chen et al. (2023) proposed *cascading*: using an auxiliary model to predict the accuracy of the small LLM's output. Madaan et al. (2023) and Zhang et al. (2023) used cascading in conjunction with multiple calls to the small model. Finally, Yue et al. (2024) propose a cascading approach that requires repeated calls of the small LLM for reasoning tasks; we show that this can be simplified to just looking at the most likely tokens for short-generation tasks.

Ramírez et al. (2023) showed that the margin on a knowledge-distilled model could optimise calls to the larger LLM. This work was limited to one pre-trained large LLM and a smaller fine-tuned local model. However, many practitioners and researchers may not be able to fine-tune their own models due to budget constraints. We extend these findings to pairs of both small and large pre-trained LLMs, as well as to other generation tasks including a multi-task setup.

Concurrent work (Gupta et al., 2024) showed that the uncertainty at the token level can be used to effectively leverage a smaller LLM in a cascade setup. Their focus is on the combination of the multiple tokens involved in longer generation, and propose a supervised method; we propose using the margin, a simple rule that optimises short-generation tasks without needing additional data. In addition, we show its applicability in the multi-task setup.

Our proposed method differs from previous studies as it does not require previously annotated data for the task and does not require repeated calls to the small LLM. Finally, it does not require training and deploying an auxiliary model.

## 3   Optimisation of LLM calls

### 3.1   Problem definition

In this work, we predict mappings between elements in the input space, $\mathcal{X}$, and the corresponding labels in the output space, $\mathcal{Y}$. We have access to the small and the large LLMs that we can prompt to become predictors $f_s, f_l : \mathcal{X} \to \Delta(\mathcal{Y})$, where $\Delta(\mathcal{Y})$ denotes the class of probability distributions over $\mathcal{Y}$. We simulate the online setting, where users send queries sequentially. We have $q$ queries $(x_1, \ldots, x_q) \overset{\text{iid}}{\sim} \mathcal{X}$ and we predict the corresponding labels $(y_1, \ldots, y_q)$. For each incoming query $x_i$, we decide whether to call an LLM based on a calling strategy (see below), and incur a given cost $c_s(x_i)$ or $c_l(x_i)$ respectively. The average

cost of the queries by both models is then given by $\hat{c}_\text{s} = \frac{1}{q} \sum_i c_\text{s}(x_i)$; $\hat{c}_\text{l} = \frac{1}{q} \sum_i c_\text{l}(x_i)$. We assume that $\hat{c}_\text{s} < \hat{c}_\text{l}$.

## 3.2 LLM Calling Strategies

For strategies that require training an additional model, we use a train dataset $X_\text{train}, Y_\text{train}$, and use either the corresponding labels from the small LLM only, $f_\text{s}(X_\text{train})$, or from both LLMs, $f_\text{s}(X_\text{train})$ and $f_\text{l}(X_\text{train})$, depending on the strategy. For the training of the auxiliary models, we follow the original papers as much as possible and perform a hyperparameter search where values are omitted. See Appendix A for a detailed explanation of the training process.

### 3.2.1 Routing Strategies

Since only the small or the large LLM is called in routing strategies, then for a given target average cost per query $c$ ($\hat{c}_\text{s} \leq c \leq \hat{c}_\text{l}$), we call the large LLM with a probability $p_\text{r}$ calculated from the re-arranged form of Equation 1.

$$c = (1 - p_\text{r})\hat{c}_\text{s} + p_\text{r}\hat{c}_\text{l} \tag{1}$$

**Random routing**  For every incoming query we call the large LLM with the probability $p_\text{r}$.

**Routing (Sakota et al., 2023; Lu et al., 2023)**  We train a meta-model to predict the performance of the small LLM only, given an incoming query. If this prediction is below a threshold value related to the probability $p_\text{r}$, it indicates the small LLM's performance is insufficient and thus we must call the large LLM.

**HybridLLM (Ding et al., 2024)**  The performance of both the small and large LLMs are modelled in this strategy. We train a meta-model to predict if an incoming query is likely to be better solved by the small LLM than by the large LLM. As in Routing above, if this prediction is below a threshold value related to the probability $p_\text{r}$, we call the large LLM, otherwise the small LLM.

### 3.2.2 Cascading

Since the small LLM is always called, then for a given target average cost per query, $c$, we call the large LLM with a probability $p_\text{c}$ calculated from the re-arranged form of Equation 2.

$$c = \hat{c}_\text{s} + p_\text{c}\hat{c}_\text{l} \tag{2}$$

**FrugalGPT (Chen et al., 2023)**  We train a model that, given a query and a candidate answer, predicts if the latter is correct. If this prediction is below a threshold value related to the probability $p_\text{c}$, we call the large LLM.

**Margin Sampling (ours)**  We suggest using the uncertainty of the output, namely the margin (Scheffer et al., 2001; Luo et al., 2004), defined by:

$$\text{Margin}_{f_\text{s}}(x_i) = P_{f_\text{s}}(y_i = k_1^{t=1} \mid x_i) - P_{f_\text{s}}(y_i = k_2^{t=1} \mid x_i) \tag{3}$$

where $k_1^{t=1}$ and $k_2^{t=1}$ are the first and second most likely tokens, respectively, according to the distribution of $f_\text{s}$ for the first predicted token position, $t = 1$. One advantage of this approach is that it does not require generating a full sequence to be able to compute uncertainty. Moreover, for the tasks we consider there is generally more uncertainty in the first token. If the margin is below a threshold value related to the probability $p_\text{c}$, we call the large LLM.

### 3.3 Dynamic threshold

All the investigated strategies require setting a threshold for the decision criterion, and we select a dynamic threshold in this work. An initial threshold is calculated using the first 20 queries. We do not evaluate whether to call the large LLM for these 20 queries, we only obtain outputs from the auxiliary models, or the margin value for Margin Sampling. This may require calling the small LLM depending on the strategy. We then use this distribution to calculate an initial $p_r$ or $p_c$-th percentile value. For all subsequent queries, the decision to call the large LLM is made, and the threshold is dynamically updated based on all past queries. We test the effect of the dynamic threshold in additional experiments (Table 10) and conclude it does not result in major performance inefficiencies.

## 4 Experimental setup

### 4.1 LLMs

We study three pairs of small and large LLMs in our experiments: Mistral 7B (Jiang et al., 2023) and Mixtral 8x7B (Jiang et al., 2024); Llama-2 of size 13B and Llama-2 of size 70B (Touvron et al., 2023); GPT-3 (Brown et al., 2020) and GPT-4 (OpenAI, 2023). We selected these pre-trained LLMs because of their popularity among practitioners, as well as to show robustness across different scales. We have arranged the pairs this way to keep the family of LLMs similar; however, this is not a requirement for any of the calling strategies.

For the open-source families (Mistral, Llama-2), all our experiments are done locally in one NVIDIA A100 GPU (80 GB), after applying a 4-bit quantisation. Section C contains more details about the LLMs used.

### 4.2 Datasets

We draw inspiration from Liang et al. (2022) and Ramírez et al. (2023) and choose a wide range of tasks, showcasing different difficulties. The sizes of the datasets, along with the label distribution and accuracy of the LLMs, can be found in Appendix (Section B, Section C).

**Classification tasks**   For emotion classification we use ISEAR (Shao et al., 2015); for fact-checking we use FEVER (Thorne et al., 2018); for sentiment analysis we use RT-Polarity (Pang & Lee, 2005), CR (Ni et al., 2019), and SST-2 (Socher et al., 2013). All of these datasets are balanced.

**Multiple-choice**   We use Openbook (Mihaylov et al., 2018), a popular multiple-choice dataset that involves common knowledge of the world.

**QA - short generation**   We use NaturalQuestions (Kwiatkowski et al., 2019), that contains real questions from human users; we use Wikifact (Petroni et al., 2019; Goodrich et al., 2019), which consists of a knowledge base completion problem to test factual knowledge; we use bAbI (Weston et al., 2016), that tests language understanding and reasoning.

### 4.3 Experiment details

For each dataset we set aside $1,000$ data-points for a train dataset that is used only to train the auxiliary models. The remainder is used for the online test set. We use $n = 500$ data-points from the train dataset in the Routing, HybridLLM and FrugalGPT strategies when training the auxiliary model, unless stated otherwise. For the auxiliary models, we fine-tune DistilBERT (Sanh et al., 2019), as per Sakota et al. (2023); Ding et al. (2024); Chen et al. (2023); Madaan et al. (2023); Ding et al. (2024). We further split the training data into 80% train and 20% validation, and fine-tune for 100 epochs with early stopping and patience of 20 epochs.

Unless stated otherwise, our results assume a simple cost scheme with $c_s(x_i) = 1$ and $c_l(x_i) = 10$, consistent with the pricing of commercial APIs (Wang et al., 2024) and similar

to the cost schemes of related work (Madaan et al., 2023; Chen et al., 2023; Lu et al., 2023; Sakota et al., 2023). We do not take into account in our experiments the latency of running DistilBERT, which we deem negligible compared to running the LLMs. To evaluate accuracy across budgets, we report Area Under the Curve (AUC) of the accuracy divided by $\hat{c}_l - \hat{c}_s$. Bolded results mark best performance, and underlined results mark second-best. We run our experiments with three random seeds, and report average results.

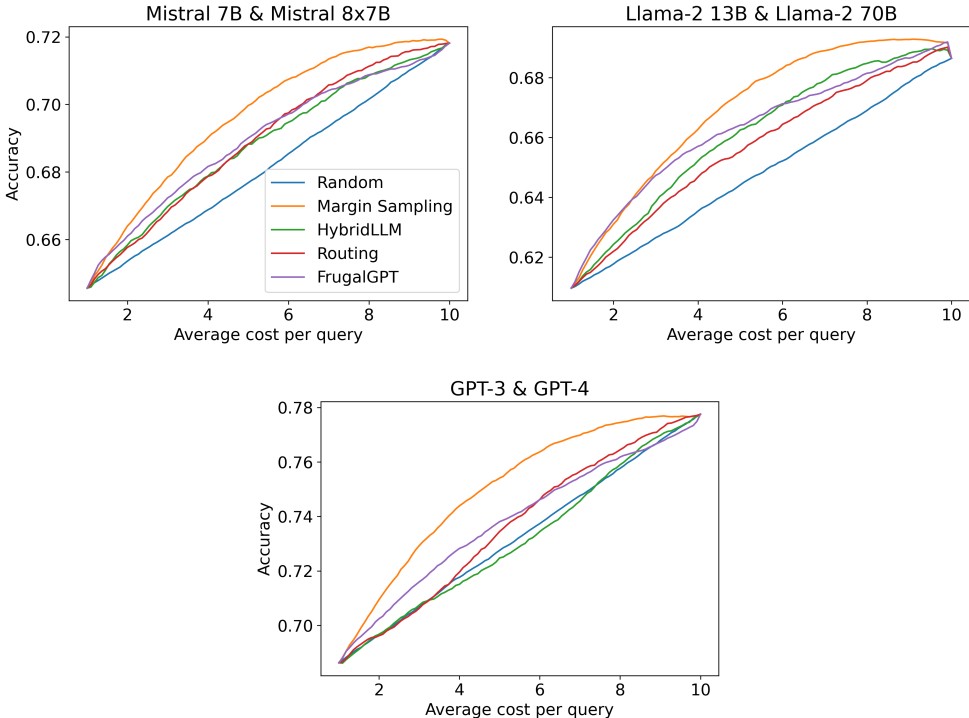

Figure 2: Accuracy curve with respect to budgets. We have averaged results for all the tasks.

## 5 Results

### 5.1 Comparison of calling strategies

Table 1 shows the AUC for all five evaluated calling strategies, across the nine tasks and for the three pairs of LLMs. Averaged across all nine tasks (the final column), we see that Margin Sampling outperforms all strategies for all LLM pairs. Across individual tasks, we see that it performs best or second best for seven of the nine tasks consistently across all three LLM pairs. Of all the nine tasks and three LLM pairs, it does not achieve best or second-best performance for only two of the 27 combinations: for ISEAR and for NaturalQuestions for the Mistral 7B and Mixtral 8x7B LLM pair. This is likely due to the poor accuracy of the small LLM Mistral 7B on these tasks (Appendix C). Figure 2 shows how performance changes with budget. We see that Margin Sampling dominates across all budgets, despite not using any training data.

Indeed, the performance of Margin Sampling seems to improve as the performance of the cheaper LLM improves, which is to be expected; it achieves its best results when applied on top of GPT-3.

FrugalGPT is on average the second-best performing strategy. This is due to its good performance on classification tasks, which is expected as it uses a classifier trained on the task as the auxiliary model. However, it performs worse than the random baseline on the more

| | ISEAR | RT-Pol | FEVER | CR | SST-2 | Openbook | Wikifact | bAbI | NaturalQ | Average |
|---|---|---|---|---|---|---|---|---|---|---|
| | | | | Mistral 7B - Mixtral 8x7B | | | | | | |
| Random | 0.606 | 0.876 | 0.773 | 0.923 | 0.880 | 0.843 | 0.443 | 0.597 | 0.193 | 0.681 |
| Routing | 0.618 | 0.876 | **0.777** | 0.924 | 0.890 | 0.844 | 0.492 | 0.606 | 0.177 | 0.689 |
| HybridLLM | 0.618 | 0.876 | 0.776 | 0.924 | 0.886 | 0.849 | 0.454 | **0.612** | **0.199** | 0.688 |
| FrugalGPT | **0.632** | **0.887** | **0.777** | 0.931 | **0.901** | 0.835 | 0.477 | 0.596 | 0.172 | 0.690 |
| Margin Sampling | 0.617 | 0.885 | **0.777** | **0.933** | 0.899 | **0.868** | **0.499** | 0.606 | 0.187 | **0.697** |
| | | | | Llama-2 13B - Llama-2 70B | | | | | | |
| Random | 0.630 | 0.809 | 0.653 | 0.885 | 0.873 | 0.617 | 0.505 | 0.600 | 0.259 | 0.648 |
| Routing | 0.639 | 0.836 | 0.662 | 0.909 | 0.883 | 0.621 | 0.514 | 0.593 | 0.254 | 0.657 |
| HybridLLM | 0.641 | 0.844 | 0.681 | 0.899 | 0.874 | 0.626 | 0.514 | 0.608 | 0.264 | 0.661 |
| FrugalGPT | **0.662** | **0.856** | 0.668 | **0.918** | **0.899** | 0.598 | 0.507 | 0.602 | 0.258 | 0.663 |
| Margin Sampling | 0.645 | 0.853 | **0.691** | 0.912 | 0.893 | **0.640** | **0.516** | **0.609** | **0.270** | **0.670** |
| | | | | GPT-3 - GPT-4 | | | | | | |
| Random | 0.747 | 0.914 | 0.816 | 0.931 | 0.898 | 0.877 | 0.552 | 0.574 | 0.281 | 0.732 |
| Routing | 0.769 | 0.915 | 0.815 | 0.936 | 0.898 | 0.878 | 0.558 | 0.584 | 0.277 | 0.737 |
| HybridLLM | 0.744 | 0.914 | 0.821 | 0.932 | 0.899 | 0.882 | 0.556 | 0.558 | 0.278 | 0.732 |
| FrugalGPT | 0.767 | 0.922 | 0.819 | **0.940** | **0.903** | 0.876 | 0.564 | 0.572 | 0.283 | 0.738 |
| Margin Sampling | **0.771** | **0.925** | **0.826** | **0.940** | 0.899 | **0.918** | **0.584** | **0.598** | **0.294** | **0.751** |

Table 1: Accuracy (AUC) for the three LLM model pairs across the five classification tasks (columns 2-6), the multiple-choice task (column 7) and the three generation tasks (columns 8-10).

challenging multiple-choice task, Openbook. FrugalGPT also performs inconsistently for QA tasks; we conclude that FrugalGPT may be satisfactory on relatively easy classification tasks and struggle with harder generation tasks.

Finally, Routing and HybridLLM seem to have a good performance in QA tasks while having a worse performance in classification tasks. We note that HybridLLM on average has the same performance as random for the OpenAI models, which is a surprising finding.

We have not shown comparisons to approaches that require repeated calls to a small LLM throughout this work, as in preliminary experiments we found they do not perform well (see Section D.2).

## 5.2 Multi-task setting

LLMs are often used to handle various tasks simultaneously. To simulate this scenario, we create an artificial multi-task setting by merging the datasets from all nine tasks. We then sample 10,000 data-points. We split this dataset into a 10% train set ($n = 1,000$ data-points) and a 90% online test set.

Figure 3 and Table 2 show the results for our multi-task experiments. We see again that Margin Sampling has the best performance. This shows the versatility of this method, that it can be applied across tasks with ease. In contrast, HybridLLM has a poor performance both for Llama-2 and OpenAI models. We found these results still hold when using 5,000 data-points as training data (Appendix D.1).

## 5.3 Robustness of experiments

### 5.3.1 Investigating the effect of training data

To ensure that auxiliary models are not being unfairly handicapped by a low-data setting, we train them with double the data points as before ($n = 1,000$), in spite of this being a possibly infeasible amount of data for many researchers and practitioners.

|                 | **Mistral** | **Llama-2** | **OpenAI** |
|-----------------|-------------|-------------|------------|
| Random          | 0.718       | 0.681       | 0.775      |
| Router          | 0.731       | 0.696       | 0.786      |
| HybridLLM       | 0.726       | 0.676       | 0.773      |
| FrugalGPT       | 0.733       | 0.694       | 0.781      |
| Margin Sampling | **0.736**   | **0.704**   | **0.792**  |

Table 2: Accuracy (AUC) in the multi-task setting. Methods Router, HybridLLM and FrugalGPT have been trained with $n = 1,000$ data-points.

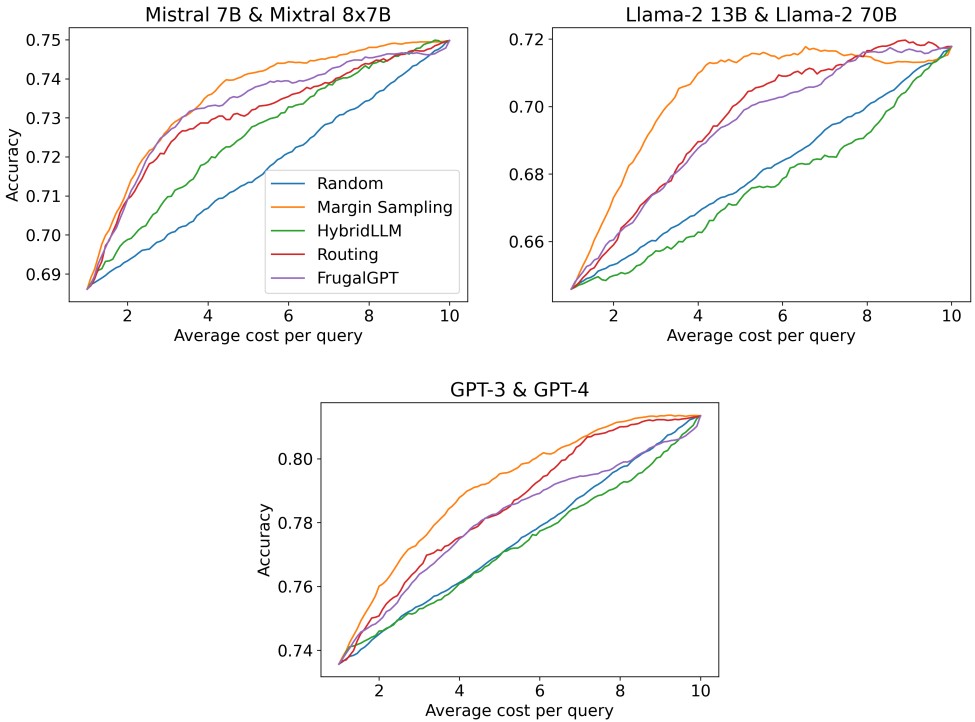

Figure 3: Accuracy curve with respect to budgets, in the multi-task setting.

Table 3 shows the averaged AUC results across tasks for the three LLM pairs. The additional data generally improves the performance of the auxiliary models, however, Margin Sampling still performs competitively, achieving the best performance for two of the three pairs. Table 3 also shows that the gap with FrugalGPT closes and that HybridLLM has again a limited performance on the OpenAI models. Additionally, we run experiments in settings with more data ($n = 5,000$), and confirm that Margin Sampling still performs competitively (Table 11).

### 5.3.2 Investigated the effect of cost

While previous studies typically only study a single cost setting (Sakota et al., 2023; Chen et al., 2023; Madaan et al., 2023; Ding et al., 2024), this could influence findings. We carry out further experiments with alternative cost schemes for the three LLM pairs across all tasks. [2] The cost ratio $r_{\text{cost}} = c_l/c_s$ is sufficient to parameterise the effect of cost. We need only vary $c_l$, and maintain $c_s = 1$. We study values of $c_l = 2, 5$ and $20$.

---

[2] We note that, at the time of writing, the pricing ratio for the OpenAI models that we used (GPT-3 and GPT-4) is $\frac{\$30/\text{million tokens}}{\$2/\text{million tokens}} = 15$

|                 | Mistral   | Llama-2   | OpenAI    |
| --------------- | --------- | --------- | --------- |
| Random          | 0.681     | 0.648     | 0.732     |
| Router          | 0.694     | 0.661     | 0.741     |
| HybridLLM       | 0.691     | 0.664     | 0.734     |
| FrugalGPT       | 0.695     | **0.672** | 0.743     |
| Margin Sampling | **0.697** | 0.670     | **0.751** |

Table 3: Accuracy (AUC, averaged across tasks) when Router, HybridLLM and FrugalGPT have been trained with 1,000 data-points.

|                 | Mistral     |             |              | Llama-2     |             |              | OpenAI      |             |              |
| --------------- | ----------- | ----------- | ------------ | ----------- | ----------- | ------------ | ----------- | ----------- | ------------ |
|                 | $c_1$=2     | $c_1$=5     | $c_1$=20     | $c_1$=2     | $c_1$=5     | $c_1$=20     | $c_1$=2     | $c_1$=5     | $c_1$=20     |
| Random          | 0.681       | 0.681       | 0.681        | 0.648       | 0.648       | 0.648        | 0.732       | 0.732       | 0.732        |
| Router          | **0.690**   | 0.690       | 0.689        | 0.657       | 0.657       | 0.657        | **0.737**   | 0.737       | 0.736        |
| HybridLLM       | 0.689       | 0.688       | 0.688        | **0.661**   | 0.661       | 0.661        | 0.732       | 0.732       | 0.731        |
| FrugalGPT       | 0.677       | 0.687       | 0.691        | 0.650       | 0.660       | 0.664        | 0.721       | 0.735       | 0.740        |
| Margin Sampling | 0.683       | **0.694**   | **0.698**    | 0.654       | **0.667**   | **0.671**    | 0.734       | **0.747**   | **0.752**    |

Table 4: Accuracy (AUC, averaged across datasets) under cost schemes $c_s = 1$ and varying $c_1$.

Table 4 shows the results of these experiments. Independently of the cost, Margin Sampling appears the best cascading strategy (against FrugalGPT). In addition, we observe that Margin Sampling is the best overall strategy for $r_{cost} \geq 5$; for $r_{cost} = 2$, routing strategies could be preferred. Intuitively, cascading needs the cost of the small LLM to be relatively cheap enough to not sacrifice too many calls to the large LLM ($p_c < p_r$).

### 5.3.3 Mixing different families of LLMs

Margin Sampling can also be applied when the two LLMs are from different families; we further test its robustness with pairs Llama-2 13B - Mixtral 8x7B, Mistral 7B - Llama-2 70B. Our findings confirm the effectiveness of Margin Sampling (Table 5).

## 6 Discussion

In this paper, we have used a simple measure of confidence of the small LLM, known as Margin Sampling, to estimate the uncertainty of an LLM output in short-generation tasks, and we leave for future work the generalisation of this approach to long-generation tasks. Our experiments show that Margin Sampling performs consistently well on a range of short-generation tasks relevant to researchers and practitioners, such as QA and multiple-choice/classification tasks.

Existing approaches on the optimisation of LLM calls require training an auxiliary model. We hypothesised that these approaches introduce additional complexity that obfuscates their

|                 | Llama-2 13B - Mixtral 8x7B | Mistral 7B - Llama-2 70B |
| --------------- | -------------------------- | ------------------------ |
| Random          | 0.664                      | 0.666                    |
| Router          | 0.673                      | 0.688                    |
| HybridLLM       | 0.672                      | 0.687                    |
| FrugalGPT       | 0.678                      | 0.688                    |
| Margin Sampling | **0.685**                  | **0.690**                |

Table 5: Accuracy (AUC, averaged across datasets) when the LLMs are from different families.

understanding, and our findings that HybridLLM performs poorly with the OpenAI models lends credibility to this hypothesis. In contrast, we propose moving towards research that leverages information within the LLM itself. Classic notions of the uncertainty of the LLM's generation, such as perplexity, margin or entropy, could be relevant signals to help optimise LLM calls. A natural extension of our work is to generalise it for an arbitrary task length, for which it may be that a global notion of uncertainty also heavily depends on the first token. Our preliminary results with Machine Translation (Appendix D.5) indicate that some further adaptation of our method may be required for tasks where the first token may not be directly or partially the answer.

It was beyond the scope of this work to investigate cascading and routing strategies of three or more LLMs. However, we hypothesise our approach may still perform well, as Margin Sampling has shown to be robust to different-sized LLMs.

## 7 Conclusions

We have proposed a method for LLM call optimisation that achieves superior performance without the need of an auxiliary model. To the best of our knowledge, the field of LLM call optimisation has not yet gained widespread adoption among practitioners. This may be due to the limited versatility and increased complexity of previous solutions. In contrast, our proposed simple approach can be easily and quickly implemented with most commercial LLMs. We believe that our findings could encourage a new direction in the research of LLM call optimisations.

### Acknowledgements

We acknowledge Yumnah Mohamied for her help and discussions while writing this paper. GR is supported by the UKRI Centre for Doctoral Training in Natural Language Processing, funded by the UKRI (grant EP/S022481/1), the University of Edinburgh, School of Informatics and School of Philosophy, Psychology & Language Sciences. IT acknowledges support by the Dutch National Science Foundation (NWO Vici VI.C.212.053). AB has been supported by the European Union's Horizon Europe Research and Innovation programme under Grant Agreement No 101070631 (UTTER). The computations described in this research were performed using the Baskerville HPC service, funded by the EPSRC and UKRI (grant EP/T022221/1) and the Digital Research Infrastructure programme (EP/W032244/1). This work also used the Cirrus UK National Tier-2 HPC Service at EPCC, funded by the University of Edinburgh and EPSRC (EP/P020267/1). The project that gave rise to these results received the support of a fellowship from "la Caixa" Foundation (ID 100010434, grant code LCF/BQ/EU22/11930079).

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

# A  Implementation details

For methods Routing, HybridLLM and FrugalGPT we use Huggingface's `AutoModelForSequenceClassification` and the model `distilbert/distilbert-base-uncased`. For methods Routing, HybridLLM we set the target number of classes to two; for FrugalGPT, we set it to either the number of target classes for classification/multiple-choice problems or two for QA problems. We perform a hyperparameter search (grid search) on a validation set of 500 examples of Openbook and Wikifact, and we find that the different methods have a similar convergence. We decide using learning rate $\mu = 5 \times 10^{-4}$, training batch size $m = 16$ and weight decay $\lambda = 0.01$, which is consistent with the reported values of Sakota et al. (2023) and seems to generalise well across tasks. Ding et al. (2024), Chen et al. (2023) and Lu et al. (2023) do not report hyperparameter values.

## A.1  Training

**Routing**  We generate an answer using the small LLM. Then, we compare it to the gold label. The target class is 1 or 0 depending on the correctness of the answer from the small LLM.

- **Input**: 'Who wrote 'If There Is I Haven't Found It Yet?'
- **Output**: '0'
- Explanation: Llama-2 13B produces an incorrect answer.

**HybridLLM**  We generate an answer using the small and the large LLMs. Then, we use the first token $y_1$ of the gold answer $y$, to obtain the quality gap $H(x) = P_{\text{Small}}(y_1 \mid x) - P_{\text{Large}}(y_1 \mid x)$. Following Ding et al. (2024), we subtract the median of $H(x)$; then we map it to 1 or 0 depending on $H(x) > H_{\text{median}}$, thus leaving a binary classification problem of uniform classes.

- **Input**: 'Who wrote 'If There Is I Haven't Found It Yet?'
- **Output**: '0'
- Explanation: Llama-2 13B is less likely than Llama-2 70B to produce the right answer.

**FrugalGPT (classification and multiple-choice)**  We train DistilBERT with gold data. During inference, we generate an answer with the small LLM. The score is the probability DistilBERT associates to this class.

|                    | ISEAR | RT-Polarity | FEVER | CR   | SST-2 | Openbook | Wikifact | bAbI | NaturalQ |
| ------------------ | ----- | ----------- | ----- | ---- | ----- | -------- | -------- | ---- | -------- |
| Total datapoints   | 6132  | 8529        | 5289  | 3393 | 7000  | 5457     | 5733     | 3000 | 2876     |
| Number of classes  | 7     | 2           | 2     | 2    | 2     | 4        | QA       | QA   | QA       |

Table 6: Datasets used.

|             | ISEAR | RT-Polarity | FEVER | CR    | SST-2 | Openbook | Wikifact | bAbI  | NaturalQ |
| ----------- | ----- | ----------- | ----- | ----- | ----- | -------- | -------- | ----- | -------- |
| Mistral 7B  | 0.557 | 0.862       | 0.770 | 0.911 | 0.854 | 0.813    | 0.359    | 0.560 | 0.125    |
| Mixtral 8x7B| 0.655 | 0.889       | 0.779 | 0.936 | 0.906 | 0.875    | 0.530    | 0.634 | 0.260    |
| Llama-13b   | 0.599 | 0.798       | 0.613 | 0.902 | 0.867 | 0.556    | 0.416    | 0.525 | 0.212    |
| Llama-70b   | 0.661 | 0.820       | 0.691 | 0.871 | 0.882 | 0.681    | 0.590    | 0.676 | 0.307    |
| GPT-3       | 0.699 | 0.900       | 0.777 | 0.921 | 0.897 | 0.798    | 0.486    | 0.462 | 0.251    |
| GPT-4       | 0.796 | 0.929       | 0.852 | 0.943 | 0.899 | 0.956    | 0.622    | 0.695 | 0.324    |

Table 7: Accuracy of the LLMs in the studied tasks.

**FrugalGPT (QA)** We train a binary classifier that predicts if an answer is correct. To do so, we use as the positive class the gold labels. We generate answers with Llama-2 13B and tag them as either positive or negative class depending on whether they match the gold labels.

- **Input**: 'Who wrote 'If There Is I Haven't Found It Yet? ANSWER: Anna Funder'

- **Output**: '0'

- Explanation: Anna Funder is an incorrect answer.

## B   Datasets

Table 6 contains some statistics on the datasets used. All the classification datasets are uniformly distributed. For our experiments, we have reserved 1,000 datapoints for training the scorers (Router, HybridLLM and FrugalGPT) and used the remaining of the datasets for online inference.

## C   LLMs used

We load the open-source LLMs with Huggingface's `AutoModelForCausalLM.from_pretrained`, activating `load_in_4bit`.
We use models `meta-llama/Llama-2-13b-hf`, `meta-llama/Llama-2-70b-hf`, `mistralai/Mistral-7B-Instruct-v0.2` and `mistralai/Mixtral-8x7B-Instruct-v0.1`. We set the temperature to 0 and look at the most likely token.

For ISEAR, RT-Polarity, FEVER, OpenbookQA, SST-2 and CR, we use the prompts from Ramírez et al. (2023) (0-shot). For Wikifact, NaturalQuestions and bAbI we use the prompts from the HELM benchmark (Liang et al., 2022).

For the OpenAI models, we use `davinci-002` (GPT-3) and `gpt-4`. Annotating all the datasets has a cost of around $180.

Table 7 contains shows the accuracy of the LLMs across the different tasks.

|                 | Mistral | Llama-2 | OpenAI |
|-----------------|---------|---------|--------|
| Random          | 0.718   | 0.682   | 0.777  |
| Router          | 0.734   | 0.696   | 0.790  |
| HybridLLM       | 0.725   | 0.688   | 0.774  |
| FrugalGPT       | **0.739** | **0.706** | 0.791  |
| Margin Sampling | **0.739** | 0.705   | **0.794** |

Table 8: Accuracy (AUC) in the multiple-task setting. Methods Router, HybridLLM and FrugalGPT have been trained with $n = 5,000$ datapoints.

|           | ISEAR | RT-Polarity | FEVER | CR    | SST-2 | Openbook | Wikifact | bAbI  | NaturalQ |
|-----------|-------|-------------|-------|-------|-------|----------|----------|-------|----------|
| Random    | 0.630 | 0.809       | 0.653 | 0.885 | 0.873 | 0.617    | 0.505    | 0.600 | 0.259    |
| Committee | 0.620 | 0.797       | 0.591 | 0.886 | 0.863 | 0.587    | 0.473    | 0.538 | 0.234    |

Table 9: Accuracy (AUC) for the Committee method (Yue et al., 2024), for Llama-2 13B and Llama-2 70B.

# D Additional results

## D.1 Multi-task setup

We experiment to see how far we can get with Routing, HybridLLM and FrugalGPT with more data. We train these methods with $n = 5,000$ datapoints. We find that Margin Sampling still outperforms them in this setup (Table 8).

## D.2 Multiple calls to the LLM

We experiment with the method from Yue et al. (2024), which estimates the uncertainty of the generation by doing multiple calls to the small LLM. We make 5 calls, sampling with temperature $T = 1$. Our results (Table 9) reveal this method does badly in our setup. The reason for this relies that doing the multiple calls to the small LLM is relatively expensive in our setup with $c_{\text{Small}} = 1, c_{\text{Large}} = 10$.

## D.3 Investigating the effect of the dynamic threshold

To investigate the effect of the dynamic threshold used in our main experiments, we simulate an offline setup, where we can obtain the values for the different methods, order them and find the *best* threshold value. Table 10 reveals that these gold threshold values only lead to a slim improvement; we conclude that dynamic threshold does not result in major performance inefficiencies.

|                 | Mistral | Llama-2 | OpenAI |
|-----------------|---------|---------|--------|
| Random          | 0.681   | 0.648   | 0.732  |
| Router          | 0.690   | 0.657   | 0.737  |
| HybridLLM       | 0.689   | 0.661   | 0.732  |
| FrugalGPT       | 0.690   | 0.664   | 0.739  |
| Margin Sampling | **0.697** | **0.670** | **0.751** |

Table 10: Accuracy (AUC, averaged across tasks) for the offline setup. The slim differences to the online setup (Table 1) suggest that the dynamic threshold does not lead to inefficiencies.

|  | Openbook | Wikifact |
|---|---|---|
| Mistral 7B - Mixtral 8x7B | | |
| Random | 0.848 | 0.453 |
| Router | 0.850 | **0.538** |
| HybridLLM | 0.851 | 0.491 |
| FrugalGPT | 0.846 | 0.498 |
| Margin Sampling | **0.865** | 0.497 |
| Llama-2 13B - Llama-2 70B | | |
| Random | 0.607 | 0.506 |
| Router | 0.613 | **0.532** |
| HybridLLM | 0.602 | 0.529 |
| FrugalGPT | 0.617 | 0.515 |
| Margin Sampling | **0.632** | 0.518 |
| GPT-3 - GPT-4 | | |
| Random | 0.872 | 0.556 |
| Router | 0.875 | **0.604** |
| HybridLLM | 0.883 | 0.559 |
| FrugalGPT | 0.876 | 0.577 |
| Margin Sampling | **0.914** | 0.588 |

Table 11: Accuracy (AUC). Methods Router, HybridLLM and FrugalGPT have been trained with n = 5, 000 data-points.

### D.4 Experiments with additional data

We test the limits of our results when additional data is available for a particular task. We run experiments on OpenbookQA and Wikifact, which are two of the harder tasks, and we train the supervised methods with 5,000 datapoints. Table 11 shows the results. The trend, as expected, is that supervised methods improve the performance, but Margin Sampling still has a relevant performance.

### D.5 Longer generation (Machine Translation)

To test the effectiveness of Margin Sampling in a task that requires longer text generation, we run experiments on the Europarl Parallel Corpus (Koehn, 2005), specifically on the German-French direction. We use Llama-3.1 of size 8B and Llama-3.1 of size 70B (Dubey et al., 2024). For this experiment setup, we find that Margin Sampling achieves 21.9 BLEU score (Papineni et al., 2002), underperforming the random baseline (22.2 BLEU). We conclude that some further adaptation of our method is required for tasks where the first token may not be directly or partially the answer. Our previous results with tasks generating multiple tokens, ie. Wikifact and NaturalQuestions, did show that Margin Sampling improves over the random baseline where a significant portion of the answers started with stop words such as articles. Therefore, we hypothesise that individual token uncertainty -not constrained to the first token- may be a suitable metric for the optimisation of API calls. This is validated by the results from Fadeeva et al. (2024), which found that token uncertainty may be used for hallucination detection, and by the work from Gupta et al. (2024), who trained a neural model to aggregate the individual token uncertainties.

