# OpenReview forum: "Optimising Calls to Large Language Models with Uncertainty-Based Two-Tier Selection"
_colmweb.org/COLM/2024/Conference — COLM_

### Official Review · Reviewer_Zkq8 · 2024-05-01

**Rating:** 6
**Confidence:** 4
**Ethics Flag:** 1

**Summary:**

The paper addresses the optimization of calls to Language Model (LLMs) within the context of having access to both a smaller and larger LLM. The primary research objective is to determine when to utilize the larger LLM for queries that cannot be adequately resolved using the smaller one.

Main contribution of this work is the validation of Margin Sampling (Scheffer et al. 2001, Luo et al. 2004, Ramirez et al. 2023) as a criterion for selecting the larger LLM. Margin Sampling operates by assessing the difference in probabilities between the most likely and second most likely tokens according to the probability distribution of the smaller LLM for the first predicted token position. The paper further refines this threshold on a per-dataset basis with minimal examples.

The experimental evaluation focuses on various tasks including classification, multiple-choice selection, and short-text generation. Results from these experiments demonstrate the efficacy of margin sampling compared to several baseline approaches.

**Questions To Authors:**

1. How does the method perform on long-form tasks? A study on this limitation would be appreciated.

2. What happens when LLMs of different families are used for margin sampling? Is it still a reliable estimator for routing?

3. Since the threshold for routing is dynamic, what is the trend of accurate predictions in the beginning and the end? Does it improve with better thresholds from the previous examples? How crucial is the threshold?

4. What is the standard size of dataset used to train the routing module as used in FrugalGPT and other baselines?

**Reasons To Accept:**

1. The simplicity of the approach and its effectiveness in reducing average cost is significant. Because the method relies on the logits of the first predicted position, the cost of routing is small.

2. Experimental validity on short-form generation task shows the usefulness of the proposed approach. Multi-task finetuning also shows improved results over the baselines which is significant.

**Reasons To Reject:**

1. Lack of comparison with the related baseline such as speculative decoding (which also requires access to the token probability distribution as current method).

2. All the evaluation is done for short-form generation tasks, what happens when the method is used for long-form generation such as paraphrase, translation, etc. Even if the results are negative, the discussion on long-form generation is crucial to understand the limitation of proposed approach.

3. Limited novelty as the idea of using Margin sampling is proposed in the similar context by Ramirez et al. 2023.

---

> ### Author Rebuttal · Authors · 2024-05-31
>
> Speculative decoding is not directly relevant to operational costs-saving approaches as it involves calling the target (large) model for each query in the verification step, which results in the same performance as the random baseline.
>
> We will incorporate additional results and a discussion on long-form generation in the final version of the paper.
>
> Ramirez et al. 2023 proposed using Margin Sampling in the context of a locally finetuned model. We show that Margin Sampling is a good router across off-the-shelf LLMs and outperforms recent works that use neural routers. Moreover, we extend results for short-form generation and a multi-task setup.
>
> LLMs of different families
>
> Since Margin Sampling only uses the probability distribution of the small model, we hypothesise that it should still be a reliable estimator for routing when using LLMs of different families. To validate this, we conduct additional experiments. The following results are in the same conditions as Table 1, and we show the average across tasks.
>
> | METHOD | Small: Llama-13b Large: Mixtral 8x7b | Small: Mistral 7b Large: Llama-70b |
> |---|---|---|
> | Random | 0.664 | 0.666 |
> | Router | 0.673 | 0.688 |
> | HybridLLM | 0.672 | 0.687 |
> | FrugalGPT | 0.678 | 0.688 |
> | Margin Sampling | **0.685** | **0.690** |
>
> How crucial is the threshold?
>
> The usage of a dynamic threshold is an artefact of the online nature of the setup. In preliminary experiments, we did not observe a substantial difference between the dynamic threshold and doing an offline evaluation (i.e., assuming we have the whole dataset, therefore we can order the values and find the ‘best’ threshold values). We present here these results, which are averaged across tasks. The following should be compared with the average column in Table 1. We observe that these gold threshold values only lead to slim improvements, and conclude that dynamic threshold does not result in major performance inefficiencies.
>
> |  | Mistral | Llama-2 | OpenAI |
> |---|---:|---:|---:|
> | Random | 0.681 | 0.648 | 0.732 |
> | Router | 0.690 | 0.657 | 0.737 |
> | HybridLLM | 0.689 | 0.661 | 0.732 |
> | FrugalGPT | 0.690 | 0.664 | 0.739 |
> | Margin Sampling | **0.697** | **0.670** | **0.751** |
>
> What is the standard size of dataset used to train the routing module?
>
> FrugalGPT do not report how many examples are used for training [1]. In [2], they assume less than 1,000 data points are available per task.
>
> [1] https://arxiv.org/abs/2305.05176
>
> [2] https://arxiv.org/abs/2308.06077

---

> > ### Comment · Reviewer_Zkq8 · 2024-06-05
> >
> > Thanks for the rebuttal. I have raised my rating.

---

### Official Review · Reviewer_KTCU · 2024-05-09

**Rating:** 6
**Confidence:** 3
**Ethics Flag:** 1

**Summary:**

This paper tackles the challenge of optimizing LLM calls to two LLMs that gap in parameter size with margin sampling for a better cost-performance trade-off. Specifically, the proposed method leverages uncertainty, defined as the probability difference of the most and the second most likely tokens, of the small LLM's first predicted token position with a statistically obtained dynamic threshold to decide if a secondary call of the large LLM is necessary. The evaluation conducted on a comprehensive set of short-generation tasks and the comparison with a comprehensive set of strong baseline methods warrant the effectiveness of a margin sampling based calling strategy.

**Questions To Authors:**

- Could you provide the variance or standard deviation for the reported results?

- How would the order of samples affect the effectiveness of dynamic thresholding?

**Reasons To Accept:**

- The proposed method is simple yet effective on short-generation tasks, and it doesn't rely on auxiliary models or additional data

- Comprehensive experiments on short-generation tasks: benchmarked recent works on LLM call optimization research, conducted multi-task and robustness assessments, and reported the average three-run results

**Reasons To Reject:**

- The authors only conducted experiments on short-generation tasks citing the evaluation complexity of long-generation tasks, which leaves a common use-case of LLMs unevaluated.

- It seems that the threshold is a key factor of the performance, but this paper only conducted experiments with the dynamic thresholding strategy that relies on stats of seen samples. The paper also lacks adequate analysis and assessment of the dynamic thresholding strategy.

- This paper is a perfect fit for an industry track, but it might not be of very significant interest to the research community as evidenced by the rejection reason of the FrugalGPT paper on OpenReview. I'm leaning towards an open mind on research topic but would like to keep a note here for the meta-reviewer's reference.

---

> ### Author Rebuttal · Authors · 2024-05-31
>
> > The authors only conducted experiments on short-generation tasks citing the evaluation complexity of long-generation tasks, which leaves a common use-case of LLMs unevaluated.
>
> Short generation is a common use case of LLMs in research and practical settings, and we show that Margin Sampling is effective. We do not expect that it would work for longer text generation, where a more complex and expensive technique may be required. We will incorporate additional results and a discussion on long-form generation in the final version of the paper.
>
> > The paper also lacks adequate analysis and assessment of the dynamic thresholding strategy [...] How would the order of samples affect the effectiveness of dynamic thresholding?
>
> The usage of a dynamic threshold is an artefact of the online nature of the setup. In preliminary experiments, we did not observe a substantial difference between the dynamic threshold and doing an offline evaluation (i.e., assuming we have the whole dataset, therefore we can order the values and find the ‘best’ threshold values). We present here these results, which are averaged across tasks. The following should be compared with the average column in Table 1. We observe that these gold threshold values only lead to slim improvements, and conclude that dynamic threshold does not result in major performance inefficiencies.
>
> |  | Mistral | Llama-2 | OpenAI |
> |---|---:|---:|---:|
> | Random | 0.681 | 0.648 | 0.732 |
> | Router | 0.690 | 0.657 | 0.737 |
> | HybridLLM | 0.689 | 0.661 | 0.732 |
> | FrugalGPT | 0.690 | 0.664 | 0.739 |
> | Margin Sampling | **0.697** | **0.670** | **0.751** |
>
> Regarding how the order of samples affects dynamic thresholding, this is accounted in our experiments (by running on three seeds).
>
> > Could you provide the variance or standard deviation for the reported results?
>
> The values of the standard deviation of our experiments in general are much lower than the precision of the results that we report. The reason for this is that we report Area Under the Curve (AUC) across 100 budgets. You can find the values of the standard deviation for Table 1 at https://shorturl.at/W8aty.
>
> > This paper is a perfect fit for an industry track, but it might not be of very significant interest to the research community [...]
>
> We want to note that several of the works we compare against have been published in conferences such as ICLR and WSDM; there has been a growing scientific interest in the problem.

---

> > ### Comment · Reviewer_KTCU · 2024-06-05
> > **Response to authors' rebuttal**
> >
> > Thank you for providing additional information. I'm convinced of the method's effectiveness and the appropriateness of using dynamic sampling. I have slightly increased the rating assuming the authors will _incorporate additional results and a discussion on long-form generation in the final version of the paper_.

---

### Official Review · Reviewer_VLt4 · 2024-05-10

**Rating:** 5
**Confidence:** 4
**Ethics Flag:** 1

**Summary:**

The paper proposes a strategy for optimisation of LLM calls - a practical problem when making multiple API calls for solving a problem. There are two main strategies - a cascading strategy and a routing strategy. The strategy choice depends on a decision criterion in an auxiliary neural model.

**Questions To Authors:**

Both small and large models contain billions of parameters. What would constitute the threshold for categorising a model as small or large?

**Reasons To Accept:**

The strategies might have the potential to be useful for someone started working with LLM API calls.

**Reasons To Reject:**

The strategies can be viewed as descriptions of practices that worked for a set of pairs and might not be sufficient for a research paper.

---

> ### Author Rebuttal · Authors · 2024-05-31
>
> >The strategies can be viewed as descriptions of practices that worked for a set of pairs and might not be sufficient for a research paper.
>
> We disagree with the statement that the strategies can be viewed as descriptions of practices that worked for a set of pairs. We compared Margin Sampling against Šakota et al. (WSDM 2024), Ding et al. (ICLR 2024) and Chen et al. (ES-FoMo at ICML 2024). In addition, we compare against Yue et al (ICLR 2024) in Table 8. For the training of the auxiliary models, we follow the original papers as much as possible and perform a hyperparameter search where values are omitted.
>
> >Both small and large models contain billions of parameters. What would constitute the threshold for categorising a model as small or large?
>
> Regarding what constitutes a ‘small’ or ‘large’ model: model size has been dramatically increasing in recent years, deeming absolute thresholds as susceptible to becoming outdated. Instead, we opt to refer to ‘small’ or ‘large’ in purely relative terms, as implied in 3.1 Problem Definition. The practitioner would have access to two models of different costs, and in this paper we use ‘small’ as a synonym for ‘cheaper’. Since we don’t want to assume any size, we study a variety of parameter sizes for models (7b - Mistral, 13b - Llama, and GPT-3, rumoured to be 175b) and do ablations with multiple relative cost settings (Section 5.3.2).

---

### Official Review · Reviewer_Q76R · 2024-05-15

**Rating:** 6
**Confidence:** 3
**Ethics Flag:** 1

**Summary:**

This work aims at optimizing the LLM inference cost by running user queries through small models and switching to larger models when the performance is not satisfying.
In contrast to other work that trains router models to predict the poor small model performance, the paper proposes to use the margin token prediction as a metric to decide when to switch between small and large models.
The proposed method is compared to several supervised router approaches including Routing, FrugalGPT, Hybrid LLM, and random. The proposed margin sampling approach outperforms the other methods on most tasks.

**Questions To Authors:**

What is the maximum number of examples that can be  used for training the rankers?

**Reasons To Accept:**

- The proposed method does not require tuning a model to decide when to switch the prediction.
- The results show that the method is efficient compared to the rest of the evaluated examples.
- The method has been tested on several model families including Mistral, Llama, and OpenAI models through the API.

**Reasons To Reject:**

- The supervised models trained for comparison seem rather weak. Only 500-100 examples are used for training which does not sound optimal. Moreover, the various models are trained on top of distilbert which is a very small model. Since the training cost is negligible, it makes sense to train the compared models on more data and with a much larger model. Running BERT or Roberta for classification would be much smaller than using the actual model.
- The evaluation uses only short context tasks and it is not clear how the proposed approach would work on a longer context. One concern is that the margin sampling might not be super efficient in larger contexts.

---

> ### Author Rebuttal · Authors · 2024-05-31
>
> In the multi-task setup, we observe that the same conclusions hold when the supervised methods are trained with 5,000 datapoints (Table 7 in the Appendix).
>
> In addition, we have run experiments on OpenbookQA and Wikifact, where the supervised methods have been trained with 5,000 datapoints. The trend, as expected, is that supervised methods improve the performance, but Margin Sampling still has a relevant performance.
>
> Mistral 7B - Mixtral 8x7B
> |  | openbook | wikifact |
> |---|---:|---:|
> | Random | 0.848 | 0.453 |
> | Router | 0.850 | **0.538** |
> | HybridLLM | 0.851 | 0.491 |
> | FrugalGPT | 0.846 | 0.498 |
> | Margin Sampling | **0.865** | 0.497 |
>
> Llama-2 13B - Llama-2 70B
>
> |  | openbook | wikifact |
> |---|---:|---:|
> | Random | 0.607 | 0.506 |
> | Router | 0.613 | **0.532** |
> | HybridLLM | 0.602 | 0.529 |
> | FrugalGPT | 0.617 | 0.515 |
> | Margin Sampling | **0.632** | 0.518 |
>
> GPT-3 - GPT-4
> |  | openbook | wikifact |
> |---|---:|---:|
> | Random | 0.872 | 0.556 |
> | Router | 0.875 | **0.604** |
> | HybridLLM | 0.883 | 0.559 |
> | FrugalGPT | 0.876 | 0.577 |
> | Margin Sampling | **0.914** | 0.588 |
>
> > What is the maximum number of examples that can be used for training the rankers?
>
> There is no limit to the number of examples that can be used for training the rankers. However, our setup implicitly assumes a low-resource setting, i.e. if a practitioner has enough data, then they may directly prefer using a task-specific model like a classifier rather than using an expensive off-the-shelf LLM. In this spirit, the original papers that introduced the supervised models trained them on limited data [1-3]. For instance, [1] assumes less than 1,000 data points are available per task.
>
> While we have not explicitly studied the effects of context length, it must be noted that some tasks are generally longer. For instance, bAbI has around 250 input tokens per datapoint. The cost of Margin Sampling is independent of context length, so it would not be more expensive for longer contexts.
>
> [1] https://arxiv.org/abs/2308.06077
>
> [2] https://openreview.net/forum?id=02f3mUtqnM
>
> [3] https://arxiv.org/abs/2305.05176

---

### Decision · Program_Chairs · 2024-07-10

**Decision:**

Accept

**Comment:**

Pros:

* The proposed method aims at optimizing LLM inference calls by using a novel approach to cascading
* The paper does a thorough evaluation on several models and the results show that the method is an improvement over existing approaches.

Cons:
* Adding an analysis of long form generation would significantly improve the paper

[At least one review was discounted during the decision process due to quality]